# Novel Variants of PPP2R1A in Catalytic Subunit Binding Domain and Genotype–Phenotype Analysis in Neurodevelopmentally Delayed Patients

**DOI:** 10.3390/genes14091750

**Published:** 2023-09-01

**Authors:** Yanyan Qian, Yinmo Jiang, Ji Wang, Gang Li, Bingbing Wu, Yuanfeng Zhou, Xiu Xu, Huijun Wang

**Affiliations:** 1Center for Molecular Medicine, Pediatrics Research Institute, Children’s Hospital of Fudan University, National Children’s Medical Center, 399 Wanyuan Road, Shanghai 201102, China; yanyanqian@fudan.edu.cn (Y.Q.); 21211240008@m.fudan.edu.cn (Y.J.); tssyligang@163.com (G.L.); bingbingwu2010@163.com (B.W.); 2Neurology Department, Children’s Hospital of Fudan University, National Children’s Medical Center, 399 Wanyuan Road, Shanghai 201102, China; xiaojizi12@sina.com (J.W.); yuanfengzhou99@163.com (Y.Z.); 3Department of Child Health Care, Children’s Hospital of Fudan University, National Children’s Medical Center, 399 Wanyuan Road, Shanghai 201102, China

**Keywords:** PPP2R1A, neurodevelopmental disorders, PPP2R1A-PPP2R5D-PPP2CA complex, genotype–phenotype correlation

## Abstract

Neurodevelopmental disorders (NDDs) are a group of high-incidence rare diseases with genetic heterogeneity. PPP2R1A, the regulatory subunit of protein phosphatase 2A, is a recently discovered gene associated with NDDs. Whole/clinical exome sequencing was performed in five patients with a family with NDDs. In vitro experiments were performed to evaluate the mutants’ expression and interactions with the complex. The genotype–phenotype correlations of reported cases as well as our patients with *PPP2R1A* variants were reviewed. We reported five unrelated individuals with PPP2R1A variants, including two novel missense variants and one frameshift variant. The protein expression of the Arg498Leu variant was less than that of the wild-type protein, the frameshift variant Asn282Argfs*14 was not decreased but truncated, and these two variants impaired the interactions with endogenous PPP25RD and PPP2CA. Furthermore, we found that pathogenic variants clustered in HEAT repeats V, VI and VII, and patients with the Met180Val/Thr variants had macrocephaly, severe ID and hypotonia, but no epilepsy, whereas those with Arg258 amino acid changes had microcephaly, while a few had epilepsy or feeding problems. In this study, we reported five NDD patients with *PPP2R1A* gene variants and expanded PPP2R1A pathogenic variant spectrum. The genotype and phenotype association findings provide reminders regarding the prognostication and evidence for genetic counseling.

## 1. Introduction

Neurodevelopmental disorders (NDDs) are a group of rare diseases with high-incidence genetic heterogeneity. The genetic diagnostic rates of NDD and intellectual disability (ID) are about 30% and 40%, respectively, in the UK 100,000 genome project [1]. Protein phosphorylation is the most common post-transcriptional modification, with more than 96% occurring for serine (Ser) and threonine (Thr). Protein phosphatase 2A (PP2A) is the major phosphatase involved in dephosphorylating at least 50% of phosphorylated Ser/Thr [2]. PP2A plays a very important role in the nervous system, immune regulation, tumorigenesis, etc. [2,3,4,5,6]. The PP2A holoenzyme consists of catalytic subunits (C), scaffolding subunits (A) and regulatory subunits (B) comprising heterotrimers. With the wide application of whole exome and whole genome sequencing in clinical diagnoses, increasing numbers of PP2A subunit pathogenic variants have been identified in NDD patients [7,8,9,10,11].

In 2015, the Decipher Developmental Disorder (DDD) group reported seven patients with ID with *de novo* pathogenic variants of *PPP2R1A* and *PPP2R5D* genes [7]. Presently, diseases caused by PPP2R1A gene mutations are defined as PPP2R1A-related NDD [12], with clinical features including severe hypotonia, varying degrees of intellectual impairment and developmental delay, abnormal head circumference, epilepsy, attention deficit, dysplasia of the corpus callosum, etc. Currently, more than 60 patients have been reported [2,4,7,10,13,14,15]. These variations may change the subunit-binding feature or affect holoenzyme activity [7,10].

The *PPP2R1A* gene is located in chromosome 19q13.33, encoding PP2A scaffolding Aα subunit and a total of 589 amino acids. The gene is widely expressed and highly conserved. Aα has 15 tandem Huntington elongation A-subunit TOR (HEAT) repeats (HRs). Each HR is composed of two α-helices connected by an intra-repeat loop. These HRs form a B-subunit-binding and a C-subunit-binding domain. Currently, the reported variants of PPP2R1A have all been localized in the B-subunit-binding domain. PP2A-Aα is an all-helical structural protein that keeps the phosphatase complex together [16]. Mice with myeloid-specific deletion of Ppp2r1a, upon exposure to particular matter, showed enhanced phosphorylation of mTOR, p70S6K and 4E-BP1 [17]. Some pathogenic variants, such as Arg183Gln, also triggered hyperphosphorylation in the mTOR/p70S6K, GSK3β and Akt pathways [18].

In this study, we report five NDD patients with *PPP2R1A* gene variants, including two reported missenses, two novel missenses and one frameshift variation. We performed in vitro experiments to validate the pathogenicity of the novel variants. We also summarized the patients’ clinical features and genetic profiles, and analyzed correlations between the genotypes and phenotypes.

## 2. Materials and Methods

### 2.1. Subject and Genomic Sequencing

Genomic DNA was extracted from the peripheral blood collected from the patients and their parents using the QIAamp DNA Blood Mini kit, following the manufacturer’s instructions. The library was constructed with the Agilent SureSelect XT Human all-exon 50 Mb kit and sequenced as 150 bp paired-end runs on the Illumina X Ten platform. The sequencing was conducted following the protocols described in our published study [19]. The candidate variants detected in patients and their parents were validated via Sanger sequencing on an ABI 3500XL Genetic Analyzer (Applied Biosystems, Foster City, CA, USA).

### 2.2. In Silico Analysis

The sequence data were mapped to the human reference genome (GRCh37/hg19). Variant calling was performed using the Genome Analysis Toolkit Best Practices Pipeline (Version 3.2.2). The allelic frequencies were annotated from gnomAD (http://gnomad-sg.org/) (accessed on 12 June 2020), the ExAC database and our in-house database (more than 50,000 samples), and the variants were evaluated using SIFT (http://sift.jcvi.org/) (accessed on 24 January 2017), PolyPhen2.2 (http://genetics.bwh.harvard.edu/pph2/) (accessed on 24 January 2017), MutationTaster (http://www.mutationtaster.org/) (accessed on 24 January 2017) and CADD. Our data interpretation followed the American College of Medical Genetics and Genomics, the Association for Molecular Pathology (ACMG/AMP) guidelines and our previously published research.

### 2.3. In Vitro Functional Studies

The wild-type plasmid tagged with 3*Flag in the N terminal of PPP2R1A (EX-C0281-M12) was purchased from GeneCopoeia^TM^, and mutants (c.539T>C, c.843dupA, c.1409T>C, and c.1493G>T) were constructed with the KOD-Plus-Mutagenesis Kit (Code No. SMK-101) from Toyobo (Osaka, Japan). The constructs were sequenced to confirm that no secondary mutation was introduced. HEK293T cells were cultured in high-glucose DMEM (Gibco, Semet, NY, USA, 11995-065) supplemented with 10% fetal bovine serum (Gibco, 12483-020) and 1% (50 μg/mL) penicillin/streptomycin (Gibco, 15140-122). Once the cells had grown to 70–80% confluence, they were separately transfected with PPP2R1A^WT^ and mutants. The cells were transfected with recombinant DNA plasmids for transient expression of the proteins using Lipofectamine 3000 reagent (Invitrogen Life Technologies, Carlsbad, CA, USA) according to the manufacturer’s instructions.

For Western blot analysis, the cells were lysed with RIPA lysis buffer with 1 mM phenylmethanesulfonylfluoride (PMSF) and a protease inhibitor cocktail, both from Cell Signaling Technology (CST) (Boston, CA, USA), for 15 min on ice. The cell lysates were clarified by centrifugation at 12,000× *g* for 5 min, and the supernatant was collected and added to the LDS sample buffer. The protein extracts were separated using SDS–PAGE electrophoresis and transferred to the nitrocellulose membrane. The membrane was incubated with primary anti-Flag (CST) antibody, anti-GAPDH (CST), anti-vinculin (CST), anti-PPP2R5D (Abcam, Cambridge, UK), anti-PPP2CA (Novus, Littleton, Colorado, USA) and the secondary anti-rabbit antibody. Detection was performed using the BioRad ChemiDoc System (Version 2.4. 0.03).

Co-immunoprecipitation (Co-IP) was performed as a routine process. The HEK293T cells were transiently transfected with the mammalian expression vectors encoding Flag-PPP2R1A or variants and HA-PPP2R5D using Lipofectamine 3000. After 48 h of transfection, the cells were collected for the experiment via a cell lysis buffer with PMSF and protease inhibitor. The cell lysates were incubated with the Flag beads (Bimake, Houston, TX, USA) overnight at 4 °C with rotation. The interactions between the endogenous PPP2R5D and PPP2CA with the wild-type or mutant PPP2R1A protein were detected with Western blotting. 

### 2.4. Literature Review and Genotype–Phenotype Summary

PubMed and Web of Science were searched using the following terms: “(mutation OR variant) AND (PPP2R1A) AND (patient)”. Twenty-six publications were carefully checked, and the neurodevelopmental disorders of patients with PPP2R1A pathogenic/likely pathogenic (P/LP) variations were summarized. A search for *PPP2R1A* gene P/LP variants was also conducted using the online database (HGMD, https://my.qiagendigitalinsights.com//bbp/view/hgmd/pro/all.php/ (accessed on 1 May 2023); ClinVar, https://www.ncibi.nlm.nih.gov/clinvar/ (accessed on 1 May 2023); and Decipher, https://www.deciphergenomics.org/ (accessed on 31 January 2023)). We excluded duplicate cases.

## 3. Results

### 3.1. Clinical Report

Patient 1 was a five-month-old girl. Ultrasound showed that she had a large head circumference at five months of fetal age. The birth weight was 3250 g without any abnormalities in the delivery process. She could not lift her head and was diagnosed with hypotonia when she was five months old. The brain MRI showed brain dysplasia, dilated Virchow-Robin spaces, less white matter and a thin corpus callosum. A skeletal X-ray did not show any hip dislocation or abnormalities in the thoracolumbar spine. When she was 5 months and 22 days old, the Gesell scores of adaptive behavior, gross motor, fine motor, language development, individual and society ability were 70, 50, 68, 85 and 73, respectively. And her lower extremities showed dystrophy. At the last follow-up, when she was eight months old, her head circumference was 52 cm; however, she still could not hold her head steadily.

Patient 2 was a girl aged one year and two months. She was born at full-term gestation via cesarean delivery to a G2P1 mother. The first fetal arrest occurred in early pregnancy. Her birth weight was 3600 g. The fetal ultrasound showed bilateral ventriculomegaly and subarachnoid effusion. The brain MRI showed bilateral ventriculomegaly and brain asymmetry when she was three months old. She could raise her head unsteadily, but could not turn over at six months old; she was then diagnosed with hypotonia. She suffered from respiratory tract infection and pneumonia repeatedly. She could sit at one year old, and could say “baba, mama” at one year and five months old, at which point her head circumference was 50.5 cm.

Patient 3 was a boy aged one year and six months. He was born full-term but small for gestational age (SGA) after an uncomplicated pregnancy. He had his first seizure at seven days after birth, and epilepsy occurred four times within two months at one year and five months old. He could walk at 13 months old, and could say “baba, mama” at one year and six months old, at which point his head circumference was 45 cm. He was diagnosed with hypospadias (phalocephalus).

Patient 4 was a girl of two years and four months. She could not say any words or walk steadily. The brain MRI showed myelination delays. She had seizures two or three times per year. She was diagnosed with hyperactivity disorder. Her right-ear hearing was 40 dB.

Patient 5 was a boy of one year and three months old. He was born at term after an uncomplicated pregnancy to a G2P1 mother. His birth weight was 3300 g. He had his first seizure at ten months old, and repeated seizures from 10 months old to one year and three months old. He could walk at 11 months old, but could not say any words. The Gesell test showed that his adaptive behavior score was 10.5 months old, gross motor quotient was 14.7 months old, fine motor quotient was 11.67 months old, language developmental quotient was 8.63 months old, and individual and society quotient was 14.23 months old at one year and three months old. The brain MRI showed a large, patchy, abnormal signal shadow in the right temporo-occipital lobe and frontotemporal ventriculomegaly. He was diagnosed with Sturge–Weber syndrome, focal epilepsy and glaucoma.

### 3.2. Genetic Findings of This Study

Four missense variants (c.538A>G, p.Met180Val; c.539T>C, p.Met180Thr; c.1409T>C, p.Val470Ala; and c.1493T>C, p.Arg498Leu) and one frameshift (c.843dupA, p.Asp282Argfs*14) variant of the *PPP2R1A* (NM_014225.6) gene were identified in the patients mentioned above; four of them were confirmed to be *de novo* variants (Table 1, Figure 1A). Three variants (c.1409T>C and c.1493T>C in patients 4 and 5; c.843dupA in patient 3) have not been reported in population databases or recorded in our in-house dataset, and were all conserved among different species, including humans, chimps, rats, mice, dogs and zebrafish (Figure 1B). The potential pathogenicity of these two novel missense variants was predicted via PolyPhen 2 and MutationTaster, and were classified as “possible/probably damaged, deleterious”. The frameshift variant of c.843dupA (p.Asp282Argfs*14) was predicted to cause C-subunit-binding domain loss (Figure 1C). Two variants (c.538A>G and c.539T>C), detected in patients 1 and 2, were previously reported in three and five patients, respectively. All eight patients had macrocephaly and hypotonia, but without epilepsy. These four missense variants’ antiparallel helical structures showed no significant change (Figure 1C).

### 3.3. In Vitro Functional Assays

The Flag-PPP2R1A^WT^ and mutants (Flag-PPP2R1A^M180T^, Flag-PPP2R1A^D282Rfs*14^, Flag-PPP2R1A^V470A^, and Flag-PPP2R1A^R498L^) with HA-PPP2R5D plasmids were transfected in the HEK293T cell line to investigate the effect of the identified *PPP2R1A* variants on protein expression and binding to over-expressed PPP2R5D. Western blot results showed that the expressions of the PPP2R1A^M180T^ and PPP2R1A^V470A^ proteins were similar to the wild-type PPP2R1A, while the expression of the PPP2R1A^R498L^ protein was relatively less than the wild-type protein. The protein level of the frameshift variant PPP2R1A^D282Rfs*14^ was not decreased but showed a truncated size (from 65 kDa to 33 kDa) (Figure 2A). The Co-IP result showed Met180Thr and our three novel variants retained their binding to exogenous HA-PPP2R5D, although Met180Thr is weakly strong compared with the wild type (Figure 2A).

Then, we assessed the binding of Flag-tagged *PPP2R1A* gene variants to endogenous PPP2R5D (B’δ subunit) and PPP2CA (C subunit) by transfecting the Flag-PPP2R1A plasmids in the HEK293T cell line. A very weak band of PPP2R5D and no PPP2CA band was detected in the Arg498Leu-PPP2R1A-transfected cell lines; neither band of these two subunits was detected in the Asn282Argfs*14-PPP2R1A-transfected cell lines; a weak band of PPP2CA and a very weak band of PPP2R5D protein were detected in the Met180Thr-PPP2R1A-transfected cell lines. While no binding difference in the B’δ and C subunit was detected between the Val470Ala-PPP2R1A- and WT-PPP2R1A-transfected cell lines (Figure 2B). Our results indicated that PPP2R1A^M180T^ may partly impair the binding to the endogenous C subunit and B’δ subunit, while PPP2R1A^R498L^ and PPP2R1A^D282Rfs*14^ completely impaired the binding to both the B’δ and C subunits.

### 3.4. Genotype–Phenotype Correlation Analysis of Patients with PPP2R1A Gene Pathogenic Variations

A total of 66 patients were enrolled, including 4 patients identified in our study (the variant of c.1409T>C in patient 4 was excluded based on the function study) (Table 1 and Appendix A). Of these, 21 types of P/LP variants were identified. Arg182Trp was the most commonly reported variant, identified in 12 patients, followed by Arg258His (n = 6), Met180Thr (n = 6) and Val220Met (n = 6) (Appendix A, Figure 3A). These 21 types of variants were distributed in seven HR motifs. Except for the Thr178 variants (including Thr178Ser and Thr178Asn; each was detected in 2 patients) in the intra-loop (immediately adjacent to α helix), the other 20 variants were all located in the α helix of the HR motif (Figure 3B). The variants in 91% (60/66) of the patients were located in the HR V, HR VI and HR VII motifs (with 39/66 patient variants clustered in HR V) (Figure 3, Appendix A). 

Of the 64 patients (2 fetus cases were excluded), 25 were female and 35 were male, and 4 were unknown (4 recorded in ClinVar without sex information). The age at diagnosis ranged from neonate to 27 years old. The common features of these individuals were language delay (100%), intellectual disability/developmental delay (98.2%, 56/57), hypotonia (88%, 44/50) and motor delay (83.3%, 30/36). Among them, 54 patients had head circumference information (23 had microcephaly; 18 had macrocephaly). Interestingly, the clinical manifestation differed in patients with microcephaly and macrocephaly (Table 2). In the microcephaly group, all of the patients had brain abnormalities, and 86.4% (19/22) showed corpus callosum agenesis/hypoplasia (CCA/CCH). In the macrocephaly group, 64.3% (9/14) of the patients had brain abnormalities; 4 patients had CCA/CCH. Notably, more patients had epilepsy in the microcephaly group than in the macrocephaly group (58.8% vs. 18.75%), whereas all patients had hypotonia in the macrocephaly group and 76.5% (13/17) in the microcephaly subgroup. The detailed clinical features are shown in Table 2, with more detailed data shown in Appendix A.

We sought to examine the genotype–phenotype correlations of patients with *PPP2R1A* gene variants. The clinical features were compared in different variant sites. The variant sites were grouped into five clusters, named Thr178/Pro179 (including Thr178Ser, Thr178Asn, Pro179Leu and Pro179His), Met180 (including Met180Val, Met180Thr, Met180Lys and Met180Arg), Arg182/Arg183 (including Arg182Trp, Arg183Trp and Arg183Gln), Ser219/Val220 (including Ser219Leu and Val220Met), and Arg258 (including Arg258Ser and Arg258His). From the perspective of the patients’ head circumference, all 9 patients with the Arg258 site variants had microcephaly, and 10 of 12 (83.3%) patients with the Met180 site variants had macrocephaly (1 patient with Met180Arg had microcephaly, while 1 patient with Met180Lys had a normal head circumference) (Figure 4A, Appendix A). From the brain abnormality perspective, CCA/CCH was observed in over 50% of the patients in four clusters, except for the Met180 site variants (only 2 of 12 patients had CCA/CCH) (Figure 4B). We also compared other clinical features in the five variant clusters. We found that all five cluster patients with clinical data showed motor delays, except for two patients with Met180Val variants. All of the patients who had clinical data with Thr178/Pro179 and Met180 variants had hypotonia, but only 33.3% (2 of 6) and 9% (1 of 11) of them had epilepsy, respectively. A total of 83.3% (10/12) and 70% (7 of 10) of patients with Arg182/183 and Ser219/Val220 variants, respectively, had epilepsy. In addition, over half (83.3%, 5/6) of the patients with Arg258 site variants showed behavioral problems (3 patients did not record behavioral problems) (Figure 4C). Notably, all 10 individuals with Met180Thr/Val variants had macrocephaly and hypotonia, but none (0/9) had epilepsy. All patients who carried Phe179Leu/His, Arg182Trp, Ser219Leu and Val220Met variants were CCA/CCH (Appendix A).

## 4. Discussion

PP2A is a serine–threonine phosphatase that plays an essential regulatory role in cell signaling and physiology, including brain development and function, tumorigenesis and autoimmune diseases [2,3,18]. This phosphatase is composed of the catalytical subunit-C (encoded by PPP2CA and PPP2CB), substrate-binding regulatory subunit-B (four subunits including PP2A-B, PP2A-B’, PP2A-B’’ and PP2A-B’’’ encoded by PPP2R2, PPP2R5, PPP2R3 and PPP2R6 genes, respectively) and scaffolding subunit-A (encoded by PPP2R1A and PPP2R1B). Variations of multiple subunits of PP2A- and PP2A-related proteins have been identified in NDD patients, including PP2A core-family-subunit-encoding genes [4] such as *PPP2R1A* [10], *PPP2R5B, PPP2R5C, PPP2R5D* [8,20,21] and *PPP2CA* [9], and PP2A regulator-encoding genes like *SET, SETBP1, BOD1* and *CIP2A* [2]. However, the majority (>90%) of pathogenic variations were detected in *PPP2R1A, PPP2CA* and *PPP2R5D* [2,7,8,9,10,11,21].

PPP2R1A-related disorder is now known as Houge–Janssen syndrome 2 (MIM:616362). Currently, over 60 patients have been reported to have *PPP2R1A* gene pathogenic variations. These patients showed different clinical manifestations such as opposite abnormal head circumference types, and microcephaly and macrocephaly. To identify the correlations between the different variations of PPP2R1A with clinical features, we collected and reviewed the clinical and genetic information of 64 NDD patients (2 reported fetuses were excluded, and 4 patients in our study were included) [7,10,13,14,15,22]. We found that patients with microcephaly often had brain structure abnormalities and severe epilepsy; patients with Met180Thr/Val variants displayed macrocephaly and severe ID, but did not suffer from epilepsy; patients with Arg258His/Ser variants all had microcephaly but rarely suffered from epilepsy. We also found that the clinical features were different, even in one site with different amino acid changes. For example, patients with the Met180Thr/Val variant presented with macrocephaly, while patients with the Met180Arg had microcephaly; patients with Arg183Gln had macrocephaly, while patients with Arg183Trp had microcephaly. Different variations in PPP2R1A showed heterogeneity in the pathogenic mechanism seen in these NDD patients.

PPP2R1A variations disrupt the binding ability to the B and C subunits of PP2A [7,10]. The variations (Pro179Leu, Ser219Leu and Arg258Ser/His) localized in the B-subunit-binding domain impaired the binding capacity with overexpressed multiple B subunits and an endogenous C subunit. The pathogenic PPP2R1A variants retained their binding to overexpressed PPP2R5D [10], another PP2A-NDD disease gene (Houge–Janssen syndrome 1, MIM:616355). In our study, we found the Met180Thr variant did not significantly change the binding to exogenous PPPR5D, but partly changed the binding to endogenous PPP2R5D. The novel frameshift variation (Asp282Arg fs*14) detected in this study, loss of the C-subunit-binding domain, was also identified as impairing the binding with endogenous B’δ and C subunits. The loss of the ability to bind other PP2A subunits had an effect on the formation of a subset of PP2A holoenzymes. In addition, several known pathogenic variants (Pro179Leu, Ser219Leu, Arg182/183Trp and Arg258Ser/His) affect the activity of the p-Rb peptide, while fewer variants (like Thr178Asn, Pro179Leu and Arg258Ser/His) affected the PP2A holoenzyme [10]. The mechanism of PPP2R1A-NND causation is gain of function; however, the loss-of-function changes could not be completely excluded. A study showed that the conditional knockout of Ppp2ca in the nervous system resulted in severe microcephaly by regulating the signaling transduction of the Hippo-p73 cascade [23]. Therefore, the Asp282Arg fs*14 variant patient had microcephaly, which may be related to the loss of PP2A activity.

Additionally, variations in impaired function can cause the insufficient dephosphorylation of PP2A substrates [7,18]. Variations in Arg183Gly/Gln triggered hyperphosphorylation in the GSK3β, AKT and mTOR/p70S6K signaling pathways and increased cell growth [18], which behave in a dominant negative manner due to the gain-of-function interactions with the PP2A inhibitor TIPRL1 [18]. Indeed, some genes encoding PI3K/AKT-mTOR components also play an important role in the regulation of organ development. Pathogenic variations in PTEN, AKT1, AKT3 and PIK3CA caused a core clinical feature, increased head circumference [24,25,26]. Therefore, we can speculate that the cause of macrocephaly in patients with Arg183Gln variants may be related to abnormalities in this pathway. However, the clinical presentation of some cases is contradictory. Patients with the Arg182Trp variant showed both microcephaly and macrocephaly or normal head circumference. How these different molecular consequences in specific PPP2R1A variations requires further study. Genetically engineered animals with specific patient variations and patient-derived induced pluripotent stem cells would offer complementary opportunities for discovery and validation. 

## 5. Conclusions

Our study reported five patients with PPP2R1A gene variants and provided functional evidence; one of them was the first-reported variant in the C-subunit-binding domain. In addition, the relationship of *PPP2R1A* gene variants with patients’ clinical features was shown to have important implications for clinical follow-up and prognostic assessment.

## Figures and Tables

**Figure 1 genes-14-01750-f001:**
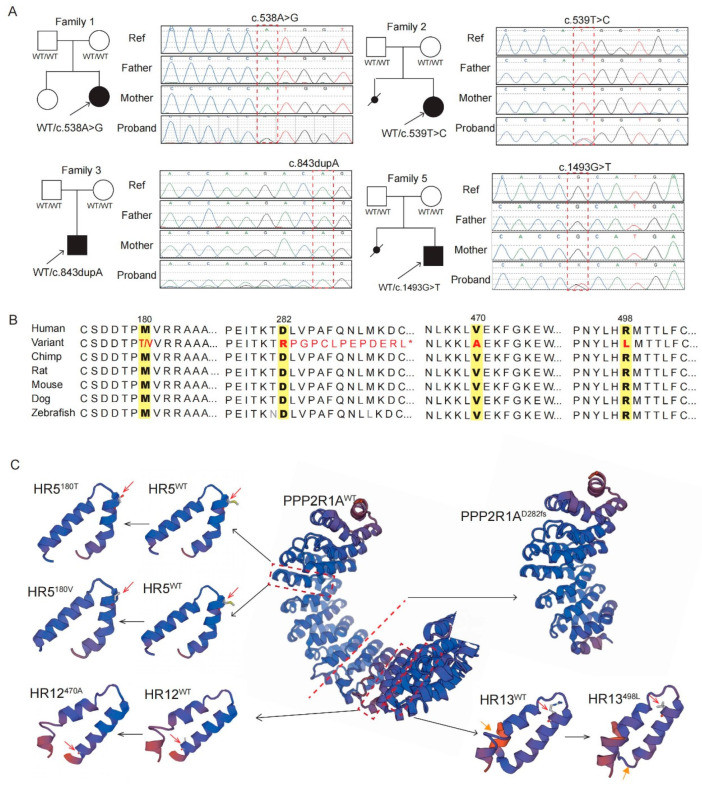
Pedigree chart, Sanger sequencing results, evolutionary conservation and HEAT repeat (HR) structure model of variants detected in our patients. (**A**) Pedigree charts of four patients with novel *PPP2R1A* gene variants and Sanger sequencing results. (**B**) Evolutionary conservation of the five variants. (**C**) Predicted senior structure of the HR antiparallel helical structures of four missense variants and the novel frameshift variant.

**Figure 2 genes-14-01750-f002:**
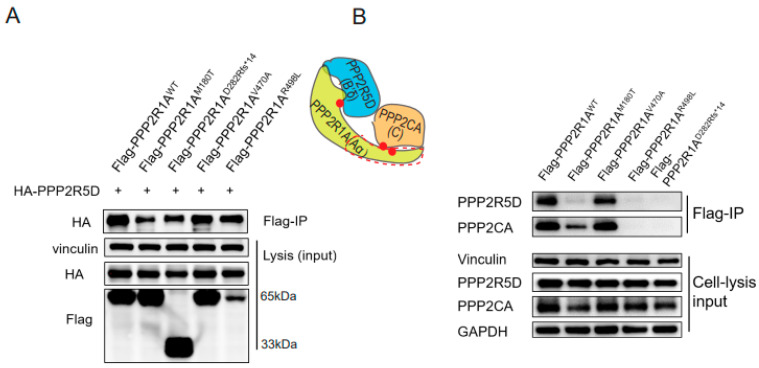
The expression of PPP2R1A and the proposed interactions of PPP2R1A with PPP2R5D and PPP2CA. (**A**) Con-transfected Flag-PPP2R1A wild-type (WT)/variants with HA-PPP2R5D in HEK293T cell to explore the ectogenic binding. The wild-type (WT) and mutant PPP2R1A protein was detected with Western blot through whole-cell lysis. Met180Thr and Val470Ala mutants’ expressions were similar to the WT protein, while Arg498Leu was relatively lower than WT and the Asn282Arg fs*14 was truncated to 33kDa. Flag pulldown con-transfected cells and Co-IP results showed that HA bands were detected in all lines, although Met180Thr-PPP2R1A was a little weak. (**B**) Flag-tagged WT-PPP2R1A and its variants were purified via flag pulldown from transfected HEK293T cells, and the presence of endogenous PPP2R5D and PPP2CA in the pulldown complexes was detected by anti-PPP2R5D and anti-PPP2CA. A weak band of PPP2CA and a very weak band of PPP2R5D were detected in Met180Thr-PPP2R1A. No bands of PPP2R5D and PPP2CA were detected in Asn282Argfs*14-PPP2R1A and Arg498Leu-PPP2R1A. No difference was detected in Val470Ala-PPP2R1A, with WT-PPP2R1A both in PPP2R5D and PPP2CA. The diagram shows the locations of the four variants and the interactions of the three subunits, the three red dots present the site of the three missense variants, the red dashed box presents the truncated region.

**Figure 3 genes-14-01750-f003:**
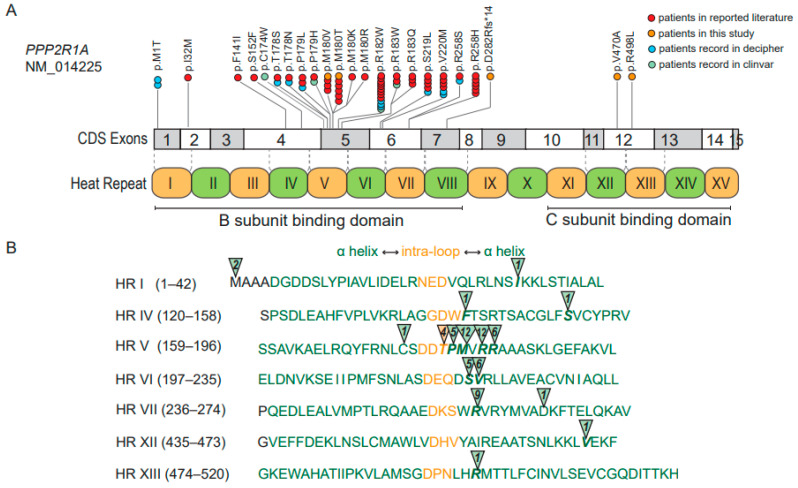
Distributions of the *PPP2R1A* gene variants were detected in 66 NDD patients. (**A**) The variant types and locations of *PPP2R1A* gene variants in exon and indications of the 15 HRs within the protein structure. Patients reported in the literature are shown by red dots, patients in this study are shown by orange dots, patients in Decipher are shown by blue dots, and patients in ClinVar are shown by green dots. Each dot represents one patient. (**B**) The number of patients and the affected amino acids are detailed in the affected HRs, with the intra-repeat loop in shown light orange and the helix in green.

**Figure 4 genes-14-01750-f004:**
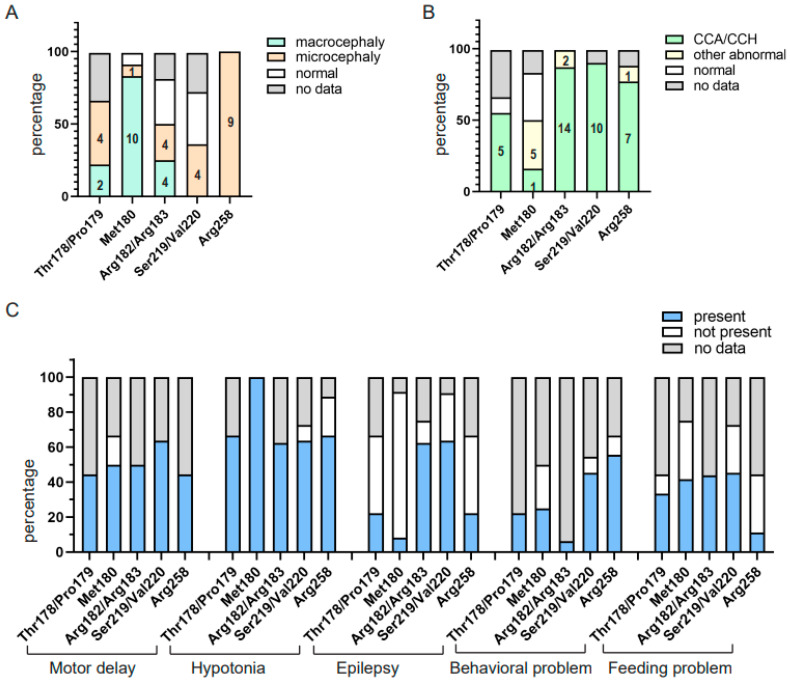
Clinical phenotypical features of patients in the five groups of *PPP2R1A* gene variants. (**A**) The comparison of the head circumference of the five groups. Patients with the Arg258 site variants all had microcephaly (n = 9), and 83.3% (10/12) of patients with the Met180 site variants had macrocephaly. (**B**) Except for the Met180 site variants, the CCA/CCH was seen in more than half of the patients in the other four groups. (**C**) Comparison of the other common clinical features in the five-variants group. Except for two patients with Met180 variants, all of the patients with clinical data with the above variants had motor delays. All patients who had clinical data with Thr178/Pro179 and Met180 variants had hypotonia, but only 33.3% and 9% had epilepsy, respectively, while most individuals with Arg182/183 and Ser219/Val220 variants (83.3% and 70%) had epilepsy.

**Table 1 genes-14-01750-t001:** The genetic evaluation and clinical features of the five patients with PPP2R1A gene variants.

ID	Patient 1	Patient 2	Patient 3	Patient 4	Patient 5
Sex/Age	F/8m	F/1y2m	M/1y6m	F/2y4m	M/1y3m
**Genotype**					
Variants (NM_014225)	c.538A>G, p.Met180Val	c.539T>C, p.Met180Thr	c.843dupA, p.Asp282Argfs*14	c.1409T>C, p.Val470Ala	c.1493G>T, p.Arg498Leu
Conservation	Y	Y	Y	Y	Y
Heat repeats	HR5	HR 5	HR 8	HR 12	HR 13
ExAC|Gnomad|in-house	0|0|0	0|0|0	0|0|0	0|0|0	0|0|0
SIFT/PP2/MT	D(0.014)/B(0.267)/D(1)	D(0)/D(0.953)/D(1)	././.	T(0.384)/P(0.535)/D(1)	D(0)/D(1)/D(1)
CADD/REVEL	23.3/0.342	25.4/0.422	./0	22.8/0.179	26.6/0.592
Class and evidence code combinations based on ACMG	Pathogenic (PS3+PS4+PM2_PP +PS2+PP2)	Pathogenic (PS2+PS3+PS4+ PM2_PP+PP2)	Likely pathogenic (PS2+PS3_P+ PM2_PP)	VUS (PM2_PP+PP2)	Likely Pathogenic (PM2_PP+PM6+ PS3_P+PP2+PP3)
Inheritance	*De novo*	*De novo*	*De novo*	NA	*De novo*
Recurrence	4 cases	6 cases	Novel	Novel	Novel
**Phenotype**					
Birth length	Normal	Normal	SGA	Normal	Normal
Macrocephaly/ microcephaly	macrocephaly	macrocephaly	microcephaly	Normal	Normal
DD/ID	DD	ID	DD	DD	DD
Language delay	No words/language	Just “baba, mama” at 1y5m	Just “baba, mama” at 1y6m	No language	No language
Motor delay	+, could not control head at 6 m	+, Sit at 1 y	-, Walk 13 m	+, Walk unsteadily	-
Behavior	NA	NA	NA	hyperactivity	Normal (at 1y3m)
Hypotonia	+	+	-	-	-
Feeding problem	-	-	+	+	-
Epilepsy	-	-	+, onset 7d, partial or GTCS	+, 2–3 times/year	+, onset 10 m
Brain MRI	Brain dysplasia, V-R space dilated, thin corpus callosum	Ventriculomegaly	Dysmyelination	Delayed myelination	Large patchy abnormal signal shadow in the right temporo-occipital lobe
Hearing loss	-	-	-	+, right ear 40 dB	-
Extremities/spine	Lower extremities dystrophy	-	-	-	-
Others			Hypospadias, phallocampsis		Sturge–Weber syndrome, glaucoma

Annotations: F, female; M, male; Y, conserved in different species; HR, heat repeat; PP2, PolyPhen2; SIFT, Sortig Intolerant From Tolerant; MT, MutationTaster; B, benign; T, tolerated; D, deleterious/probably damaging/disease causing; P, possibly damaging; CADD, Combined Annotation-Dependent Depletion; REVEL, Rare Exome Variant Ensemble Learner; VUS, variants of uncertain significance; SGA, small for gestational age; GTCS, generalized tonic–clonic seizures; V-R spaces, Virchow-Robin spaces; m, month; y, year; NA, not applicable; +, the patient presents the phenotype; -, the patient dose not present the phenotype; dB, decibel.

**Table 2 genes-14-01750-t002:** The clinical manifestations of patients with microcephaly and macrocephaly.

Phenotypic Features	Patients (Had Features/Had Records)	Percentage (%)	23 Patients with Microcephaly (Less Than −2SD)	18 Patients with Macrocephaly (Large Than +2SD)
DD/ID	56/57	98.2%	22/22	15/15
Language delay	39/39	100%	15/15	10/10
Motor delay	30/36	83.3%	12/13	9/11
Brain abnormal	47/55	85.5%	**22/22**	**9/14**
CCA/CCH	37/55	67.2%	**19/22**	**5/14**
Hypotonia	44/50	88%	**13/17**	**16/16**
Epilepsy	25/50	50%	**10/17**	**3/16**
Behavioral problem	22/27	81.5%	7/9	7/9
Feeding problem	23/38	60.5%	9/14	9/13
Heart defects	11/28	39.3%	3/8	5/10

Note: CCA, corpus callosum agenesis; CCH, corpus callosum hypoplasia; the bold numbers in the table mean they had different between microcephaly and macrocephaly group.

## Data Availability

Variants detected in this study were submitted to ClinVar with submission number SUB13097938. The raw data are not publicly available due to ethical restrictions.

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
