# Peer review of "Novel Variants of PPP2R1A in Catalytic Subunit Binding Domain and Genotype–Phenotype Analysis in Neurodevelopmentally Delayed Patients"

_genes, 2023, doi:10.3390/genes14091750_

Round 1

Reviewer 1 Report

Through whole exome sequencing studies, Qian et al. find new variants in PPP2R1A, a regulatory subunit of protein phophatase 2A. Variants in PPP2R1A have been observed previously in individuals with neurodevelopment disorders, and the variants described in this paper expand this list. Four of the variants identified were used to carry out functional studies using FLAG-tagged, over expressed, protein in HEK293T cells. Specifically, the authors examined total protein levels and the ability of mutant proteins to interact with PPP2R5D and PPP2CA by co-IP. The data presented is generally clear, although I have several concerns.

1. M180T, R498L and D282fs clearly show deficits in binding to PPP2R5D and/or PPP2CA. This is consistent with these variants being pathogenic.  This, however, is not clear for V470A, which seems to show normal interactions. It was not clear why this variant was classified as pathogenic? 

2. It was not clear how many variants (total) were detected in these patients. More transparency on the genotypic information would be nice. 

Editing for language needs to done. 

Author Response

Through whole exome sequencing studies, Qian et al. find new variants in PPP2R1A, a regulatory subunit of protein phosphatase 2A. Variants in PPP2R1A have been observed previously in individuals with neurodevelopment disorders, and the variants described in this paper expand this list. Four of the variants identified were used to carry out functional studies using FLAG-tagged, over expressed, protein in HEK293T cells. Specifically, the authors examined total protein levels and the ability of mutant proteins to interact with PPP2R5D and PPP2CA by co-IP. The data presented is generally clear, although I have several concerns.

  1. M180T, R498L and D282fs clearly show deficits in binding to PPP2R5D and/or PPP2CA. This is consistent with these variants being pathogenic. This, however, is not clear for V470A, which seems to show normal interactions. It was not clear why this variant was classified as pathogenic? 

Response: Thanks for the reviewer’s comment. We agree that the V470A showed normal interactions, and we also classified the V470A as an uncertain significance variant as in Table 1.

  1. It was not clear how many variants (total) were detected in these patients. More transparency on the genotypic information would be nice. 

Response: Patients 2 and 3 were tested trio whole exome sequencing (Trio-WES), Patients 1, 4 and 5 were tested proband-only clinical exome sequencing. Except for the variants of PPP2R1A, other genetic-causing variants were excluded in these five cases.

The de novo and compound heterozygous variants detected from Trio-WES, and the candidate variants from proband clinical exome sequencing are listed in the following table.

The variants were annotated using Variant Effect Predictor (VEP-Ensembl 73). Functional exonic and splicing variants were considered in the analysis. Allelic frequencies were annotated from public datasets, including 1000 Genomes (1KG), Exome aggregation Consortium (ExAC), GNOMAD, and our internal databases, which include over 100k samples. Further analysis of single nucleotide variants (SNV) and small indels follows an in-house pipeline, including off-target filtering, public population frequency filtering, internal population frequency filtering, variant damage prediction and inheritance model scoring. The analysis pipelines filtered out an average of ~200 variants in each sample for manual analysis. The total variants detected in these five patients were 237, 233, 164, 115 and 188, respectively. 

The de novo and compound heterozygous variants detected from Trio-WES, and the candidate variants from proband clinical exome sequencing are listed in the following table.

ID

Gene

location

Variant

Zygo

OMIM

Inherit

Source

Comments

Patient1

ANKRD11

chr16: 89357415

NM_013275:exon5: c.397+6C>T

Het

KBG syndrome, [MIM:148050]

AD

NA

1 normal individual in in-house dataset

PPP2R1A

chr19: 52715973

NM_014225:exon5: c.538A>G(p.M180V)

Het

Houge-Janssen syndrome 2 [MIM:616362]

AD

De novo

Pathogenic variant matches the patient's phenotype

PRSS12

chr4: 119234489

NM_003619:exon7:c.1356C>T(p.D452D)

Het

Intellectual developmental disorder, autosomal recessive 1, [MIM:249500]

AR

Maternal

Synonymous variant, homozygous in-house dataset without intellectual disability feature

PRSS12

chr4: 119273715

NM_003619:exon1:c.161C>G(p.P54R)

Het

Intellectual developmental disorder, autosomal recessive 1, [MIM:249500]

AR

Paternal

homozygous carrier in in-house dataset without intellectual disability feature

Patient 2

DHX58

chr17: 40259777

NM_024119:exon8: c.841_842delinsAA(p.A281K)

Het

.

De novo

Non OMIM disease gene

NUP210

chr3: 13383363

NM_024923:exon23: c.3113A>G(p.D1038G)

Het

.

Paternal

Non OMIM disease gene

NUP210

chr3:13420488

NM_024923:exon8: c.977-8A>G

Het

.

Maternal

Non OMIM disease gene

PPP2R1A

chr19:52715974

NM_014225:exon5: c.539T>C(p.M180T)

Het

Houge-Janssen syndrome 2 [MIM:616362]

AD

De novo

Pathogenic variant matches the patient's phenotype

SLC27A3

chr1:153749134

NM_001317929:exon2: c.808C>T(p.P270S)

Het

.

Maternal

Non OMIM disease gene

SLC27A3

chr1:153747984

NM_001317929:exon1: c.11_23delinsT (p.P8_L11del)

Het

.

Paternal

Non OMIM disease gene

Patient 3

AMER1

chrX:63412601

NM_152424:exon2: c.566A>G(p.Q189R)

Hemi

Osteopathia striata with cranial sclerosis, [MIM:300373]

XLD

Maternal

From normal mother

MYO5B

chr18:47431204

NM_001080467:exon20:c.2415-6C>G

Het

Microvillus inclusion disease, [MIM:251850]

AR

Maternal

Phenotype not match

MYO5B

chr18:47429121

NM_001080467:exon21:c.2654A>G(p.D885G)

Het

Microvillus inclusion disease, [MIM:251850]

AR

Paternal

Phenotype not match

PPP2R1A

chr19:52719066

NM_014225:exon7: c.842delinsCA (p.D282Rfs*14)

Het

Houge-Janssen syndrome 2 [MIM:616362]

AD

De novo

Likely pathogenic variant matches the patient's phenotype

Patient 4

ARHGAP31

chr3:119134635

NM_020754:exon12: c.3859A>G(p.T1287A)

Het

Adams-Oliver syndrome 1, [MIM:100300]

AD

NA

 6 individuals without intellectual disability were in the in-house dataset

CHRNA4

chr20:61981756

NM_000744:exon5: c.1007G>A(p.R336H)

Het

Epilepsy, nocturnal frontal lobe, 1, [MIM:600513]

AD

NA

Phenotype not match

COL2A1

chr12:48369765

NM_001844:exon50: c.3578C>T(p.S1193L)

Het

Achondrogenesis, type II or hypochondrogenesis, [MIM:200610]; [MIM:156550]; Stickler syndrome, type I, [MIM:108300]; et al

AD

NA

Phenotype not match

GJB2

chr13:20763612

NM_004004:exon2: c.109G>A(p.V37I)

Het

Deafness, autosomal dominant 3A, [MIM:601544]; Deafness, autosomal recessive 1A, [MIM:220290]; et al

AR

NA

Likely pathogenic variant, carrier

HARS1

chr5:140056390

NM_002109:exon10: c.1043G>A

Het

Charcot-Marie-Tooth disease, axonal, type 2W, [MIM:616625]; Usher syndrome type 3B, [MIM:614504]

AD/AR

NA

Phenotype not match

HTT

chr4:3208331

NM_002111:exon43: c.5827G>A

Het

Huntington disease, [MIM:143100]; Lopes-Maciel-Rodan syndrome, [MIM:617435]

AD/AR

NA

Phenotype not match

PPP2R1A

chr19:52724277

NM_014225:exon12: c.1409T>C(p.V470A)

Het

Houge-Janssen syndrome 2  [MIM:616362]

AD

NA

Uncertain significance variant matches the patient's phenotype

VCAN

chr5:82837289

NM_004385:exon8: c.8467C>G(p.Q2823E)

Het

Wagner syndrome 1, [MIM:143200]

AD

NA

Phenotype not match

Patient 5

DDX41

chr5:176938856

NM_016222:exon17: c.1786_1804del (p.P596Rfs*)

Het

{Myeloproliferative/lymphoproliferative neoplasms, familial (multiple types), susceptibility to}, [MIM:616871]

AD

NA

Phenotype not match

GJB4

chr1:35227495

NM_153212:exon2: c.640A>G(p.M214V)

Het

Erythrokeratodermia variabilis et progressiva 2, [MIM:617524]

AD

NA

Phenotype not match

PPP2R1A

chr19:52724361

NM_014225:exon12: c.1493G>T(p.R498L)

Het

Houge-Janssen syndrome 2 [MIM:616362]

AD

De novo

Likely pathogenic variant matches the patient's phenotype

RHOBTB2

chr8:22864996

NM_001160036:exon7: c.1304T>C(p.V435A)

Het

Developmental and epileptic encephalopathy 64, [MIM:618004]

AD

NA

9 individuals without intellectual disability in the in-house dataset

TRIO

chr5:14498209

NM_007118:exon52: c.8059T>A(p.F2687I)

Het

Intellectual developmental disorder, autosomal dominant 44, with microcephaly, [MIM:617061]; Intellectual developmental disorder, autosomal dominant 63, with macrocephaly, [MIM:618825]

AD

Paternal

From no feature father

3. Comments on the Quality of English Language

Editing for language needs to done. 

Response: Thanks for the reviewer’s comment. We had sent our manuscript for English languange editing.

Reviewer 2 Report

Qian et al. report yet unreported three PPP2R1A variants with functional tests and make a molecular and clinical review of the literature. However, I do not think that this work adds significant material to what is already known. Indeed, of the three novel variants, to my mind only the c.1493G>T, p.Arg498Leu seems relevant. It may be classified ACMG class 4 as the authors propose but I would use PM2 moderate, PP2 supporting, and PS2 strong arguments. The issue is that there are three pathogenic missense variants clusters and that this variant is not localized in any of them, in the 3’ side of the protein where several missense variants are reported in gnomAD. However, missense 3D predictions are in favor of a deleterious impact (Structural damage detected, Buried charge replaced Buried salt bridge breakage) but it may act as a loss of function mechanism whereas the known mechanism is dominant-negative type of gain-of-function effect (see GeneReviews).

The c.1409T>C, p.Val470Ala is clearly a class 3 variant (PM2 moderate, and PP2 supporting arguments ; authors use PP3 argument but in silico predictions are clearly not in favor of a deleterious impact). Missense 3D predictions are: No structural damage detected.

Last, the c.843dupA, p.Asp282Argfs*14 is a truncating variant. ACMG classification could be class 4 (PM2 moderate, and PS2 strong). PVS1 can not be used here because loss of function is not a known mechanism, see GeneReviews ("It should be noted that PPP2R1A loss-of-function changes (e.g., nonsense variants or deletions) are not known to cause PPP2R1A-NDD, even though the gene is quite intolerant to loss-of-function changes (gnomAD pLI is 0.98)."). To my mind, this variant is rather a class 3 variant. Besides, authors do not explain why functional tests are not in favor of NMD which should occur because of the localization of the variant.

Functional tests are not in accordance of what is known. In Figure 2B, it seems that well known pathogenic M180T variant impacts PPP2R5D binding which should not be the case, see GeneReviews (“The most important B-subunit for brain function appears to be the PP2A B56-delta subunit PPP2R5D, and pathogenic missense variants in this gene are the cause of a phenotypically overlapping condition, PPP2R5D-related neurodevelopmental disorder. Remarkably, all pathogenic PPP2R1A variants retain binding to PPP2R5D [Lenaerts et al 2021]”).

I also think that the clinical review does not add significant information compared to GeneReviews.

Reviewer 3 Report

Qian et al. identified the Novel variants in PPP2R1A and also compared the genotype-phenotype relations in NDD patients having PPP2R1A variants. Overall, the study is well-written and concise.

1. Did the author check novel variants in "Bravo" and "all of US" genome browsers?

2. What was the cut-off value for allelic frequency during exome analysis?

3. Did the author check compound het variants in exome data?

4. Are all the variants mentioned in Supplementary Table 1, de novo? if yes please mention it in the table or legend.

Round 2

Reviewer 2 Report

The authors have responded to my comments and questionings.